# Inspiratory Muscle Training Improved Cardiorespiratory Performance in Patients Undergoing Open Heart Surgery: A Randomized Controlled Trial

**DOI:** 10.3390/arm93030010

**Published:** 2025-05-27

**Authors:** Chitima Kulchanarat, Suphannee Choeirod, Supattra Thadatheerapat, Dusarkorn Piathip, Opas Satdhabudha, Kornanong Yuenyongchaiwat

**Affiliations:** 1Department of Physical Therapy, Thammasat University Hospital, Pathum Thani 12120, Thailand; pkpt.tuh@gmail.com (S.C.); ashc1802@gmail.com (S.T.); 2Cardiovascular Thoracic Intensive Care Unit (ICU CVT) Unit, Thammasat University Hospital, Pathum Thani 12120, Thailand; skornthip@gmail.com; 3Division of Cardiovascular Thoracic Surgery, Department of Surgery, Thammasat University Hospital, Pathum Thani 12120, Thailand; opascts@hotmail.com; 4Department of Physiotherapy, Faculty of Allied Health Sciences, Thammasat University, Pathum Thani 12120, Thailand; kornanong.y@allied.tu.ac.th; 5Thammasat University Research Unit, Physical Therapy in Respiratory and Cardiovascular Systems, Thammasat University Pathum, Thani 12120, Thailand

**Keywords:** inspiratory muscle strength, open heart surgery, inspiratory muscle training, cardiorespiratory performance, inspiratory training device, physical therapy program

## Abstract

**Highlights:**

**What are the main findings?**
Inspiratory muscle training effectively improved inspiratory muscle strength and cardiorespiratory performance in patients undergoing open heart surgery.

**What is the implication of the main finding?**
Inspiratory muscle training can be performed using maximum pressure resistors such as the TU-Breath Trainer devices and is an important strategy for inspiratory muscle strength and cardiorespiratory performance in patients undergoing open heart surgery.

**Abstract:**

Aim: This study aimed to evaluate the effects of inspiratory muscle training on inspiratory muscle strength and cardiorespiratory performance in patients undergoing open heart surgery. Method: This study was conducted as a randomized controlled trial with two groups. Fifty-eight patients who underwent open heart surgery were randomly assigned to either the intervention group or the control group 29 in the control group and 29 in the intervention group. Patients in the intervention group participated in a physical therapy program combined with inspiratory muscle training using the Thammasat University (TU) Breath Trainer. Patients in the control group received only the standard physical therapy program. The maximum inspiratory pressure, maximum expiratory pressure and 6 min walk test distance were assessed both before surgery and prior to hospital discharge. Results: The intervention group had a significant increase in maximum inspiratory pressure (*p* < 0.001), maximum expiratory pressure (*p* < 0.001) and 6 min walk test distance (*p* = 0.013). The control group had a significant decrease in maximum inspiratory pressure (*p* < 0.001), maximum expiratory pressure (*p* = 0.002) and 6 min walk test distance (*p* < 0.001). Conclusions: Inspiratory muscle training can be performed using maximum pressure resistors, such as the TU-Breath Trainer device. This training has been shown to effectively improve inspiratory muscle strength and cardiorespiratory performance in patients undergoing open heart surgery, as well as reduce pulmonary complications and shorten the length of hospital stay.

## 1. Introduction

Coronary artery bypass grafting (CABG) and heart valve surgery are among the most commonly performed cardiac surgical procedures worldwide [1]. CABG is used to treat myocardial perfusion in coronary artery disease (CAD). CAD occurs when the coronary arteries, responsible for supplying blood and oxygen to the heart muscle, become narrowed or blocked—typically due to the buildup of plaque, a mixture of cholesterol, fat, calcium and other substances. This leads to abnormal heart rhythms, heart failure, chest pain and shortness of breath [2,3,4,5].

In cardiac surgery, the use of anesthetics, narcotic drugs and the temporary discontinuation of ventilation during cardiopulmonary bypass (CPB) are associated with post-surgical pain, which can lead to reduced ciliary function, respiratory muscle dysfunction and impaired cough effectiveness. CPB during cardiac surgery can induce oxidative stress, resulting in pulmonary ischemia–reperfusion injury, which, in turn, triggers lung damage and a systemic inflammatory response [6,7]. This respiratory dysfunction contributes to pulmonary complications such as atelectasis, prolonged mechanical ventilation, pneumonia and pulmonary congestion, as well as systemic complications including infections, perioperative stroke, upper gastrointestinal bleeding, diabetes mellitus, renal failure and hemodynamic instability. Consequently, this results in reduced muscle strength, diminished functional capacity, impaired insulin sensitivity and increased mortality [8,9].

Inspiratory muscle training (IMT) is one of the procedures used to promote maximum inspiratory pressure (MIP), maximum expiratory pressure (MEP) and cardiorespiratory performance [10]. The recommendation for physiotherapy treatment consists of breathing techniques, coughing techniques, incentive spirometry, continuous positive airway pressure, intermittent positive pressure, positioning, early mobilization and IMT [10,11]. Several studies have reported that IMT is effective in the recovery of MIP, MEP and cardiorespiratory performance. For example, Cordeiro et al. [9] the intervention group received an IMT program at 40% of the MIP, performing three sets with 10 repetitions. They revealed that IMT led to improved inspiratory muscle strengthening (IMS) and enhanced walking distance, which also resulted in increased maximum inspiratory pressure and 6 min walk test scores. Various IMT methods, techniques, devices and protocols are available. Inspiratory pressure threshold loading such as breathing via a device is the most widely used method for improving IMS [12,13,14,15,16]. Based on the results of previous studies, IMS can be increased by using a Thammasat University (TU)-Breath Trainer, through which users breathe while it is attached to the lower costal margin [17]. Therefore, this study aimed to examine the effects of IMT on the inspiratory muscles and cardiorespiratory performance of patients following open heart surgery.

## 2. Materials and Methods

### 2.1. The Study Design and Participants

This study was approved by the Human Research Ethics Committee of Thammasat University Hospital, based on the Declaration of Helsinki, the Belmont Report, the CIOMS Guidelines and the International Conference on Harmonization Good Clinical Practice (ICHGCP) COA No. 013/2565. The CONSORT 2010 guidelines for randomized controlled trials were followed. All patients demonstrated comprehensive understanding of the study’s purpose and procedures, and informed consent was obtained from each patient who agreed to participate. This study was designed as a prospective, single-blind, randomized controlled trial to be conducted at Thammasat University Hospital.

The number of patients required for the study was calculated according to Cordeiro et al. [9] with a statistical G*power of 0.8. The total sample size was determined to be 52 patients divided into two groups: 26 patients in each group and 8 patients in case of drop-outs. Those who met the inclusion criteria for this study were (1) patients aged 35–80 years, (2) both men and women, (3) patients who were scheduled to undergo cardiac surgery (CABG, mitral valve, aortic valve and atrial septal defect correction). Patients were excluded from the study if they met any of the following criteria: (1) patients who were unable to understand the techniques, (2) history of neuromuscular disorders (3) psychiatric problems, (4) thoracotomy, (5) unstable angina, (6) uncontrolled cardiac arrhythmia, (7) hemodynamic instability, (8) high resting heart rate (≥120 beats per minute), (9) uncontrollable blood pressure or blood glucose. Prior to commencing the training program, participants received a detailed explanation of the study’s objectives and methodology. They were then asked to provide informed consent and complete a questionnaire addressing their medical history, comorbidities and tobacco use history. The lottery method was used to randomly divide patients into two groups (control and intervention).

### 2.2. Outcomes and Measures

The study focused on IMS as the primary outcome measure and used the maximal inspiratory pressure and maximal expiratory pressure as secondary outcome measures to assess functional capacity. The IMS was measured using a respiratory pressure meter (RPM 01, Micro Medical Ltd is headquartered in Rochester, Kent, ME1 2AZ, United Kingdom). The MIP was measured during the maximum inspiratory effort from the residual volume and the MEP was measured during the maximum expiratory effort from the total lung capacity. Measurements were taken with each participant seated on a chair and wearing a nose clip. Patients were asked to breathe as much as possible for at least 1 s. The evaluation was repeated until no further improvements were obtained, and three satisfactory attempts that differed by <20% were used for the analysis. The highest value was recorded in centimeters of water (cmH_2_O). The American Thoracic Society/European Respiratory Society recommends a standard testing protocol for IMS [18]. Patients who underwent open heart surgery (OHS) were asked to perform IMS before the heart operation and before discharge.

Cardiorespiratory performance was measured using the 6-MWT. The patients were instructed to walk as fast as possible along a 30 m straight, flat hospital corridor for 6 min. The total distance walked was measured to the nearest meter and recorded. Before and after the test, all patients were monitored for resting heart rate, respiratory rate, peripheral oxygen saturation and blood pressure in the sitting position [19]. The test was discontinued in the event of dizziness, palpitation or a sudden change in vital signs.

### 2.3. Procedures

Two physiotherapists recruited the patients, who were admitted to the department of cardiothoracic surgery, Thammasat University Hospital. The physical therapist had over five years of experience in providing care to patients recovering from cardiac surgery. All patients were assessed by the physical therapist both preoperatively and at discharge. All patients received the standard cardiac rehabilitation program pre- and postoperatively (i.e., cardiac rehabilitation program phase I). All patients received general anesthesia and underwent median sternotomy. Patients were provided with supplemental oxygen to maintain arterial oxygen saturation above 95% following extubation. IMT was performed using a TU-Breath Trainer device. This device, a newly developed version III of a prototype respiratory muscle training system, was employed in the present study. The system comprised an airway-resistance-based inspiratory training component and an integrated inspiration sensor. A pressure sensor was incorporated into the respiration sensor, and the measured pressure values were transmitted to a smartphone via Bluetooth. Detailed descriptions of the pressure sensor and the overall system have been provided in previous publications [17]. Patients wore a device attached to a chest strap at the lower costal margin (see Figure 1). All the patients were asked to sit on a chair and take a deep breath for 10 min, performing three sets of 15 repetitions. The intervention group (IG) performed this routine twice daily until hospital discharge, in accordance with the study protocol. The control group (CG) continued with the cardiac rehabilitation program during phase I. The principle of this device has been described in detail elsewhere [17].

### 2.4. Statistical Analysis

To determine the sample size for the study, power analysis was conducted using the G*Power program with reference to a similar article [9]. The collected data were analyzed using the SPSS statistical software version 20. The normality of distribution was assessed using the Kolmogorov–Smirnov test. Descriptive statistics were used to explain the features of the sample population and are reported as frequencies and percentages. Chi-squared tests were used to compare qualitative variables. MIP, MEP and 6MWT results were analyzed using factorial analysis of variance (ANOVA). A *p*-value of less than 0.05 was considered statistically significant.

## 3. Results

### 3.1. Participants

In total, 60 patients were enrolled in the study, and 3 patients did not complete the study due to cardiac arrest post-surgery. A total of 58 patients were included in the analysis, 29 in the IG and 29 in the CG. Forty patients underwent CABG, while twenty patients underwent valve surgery. There was a statistically significant difference between the groups in terms of surgery duration (IG: 323.67 ± 90.4 min vs. CG: 390 ± 137.6 min, *p* = 0.04), duration of mechanical ventilation (IG: 720.9 ± 274.3 min vs. CG: 891.4 ± 610.5 min, *p* = 0.04) and length of hospital stay (IG: 8.73 ± 3.69 days vs. CG: 11.7 ± 6.9 days, *p* = 0.01) (Table 1). Also, there were no significant differences in baseline assessment regarding demographic characteristics, risk factors and left ventricular ejection fraction.

### 3.2. Outcomes

Patients in the IG participated in a physical therapy program combined with IMT using a TU-Breath Trainer device, whereas those in the CG received only the physical therapy program. Significant improvements were observed in IMS (*p* < 0.001) in the IG after treatment compared to before treatment (Figure 2). Similarly, the IG showed significant improvements in the 6MWT (*p* < 0.001) after treatment compared to before treatment (Figure 3). The radiological diagnostic reports from the postoperative days were taken into consideration. In this study, there was a statistically significant difference between groups (*p* < 0.001). Regarding the presence of atelectasis in the CG, 11 patients (36.7%) were affected (Table 2).

## 4. Discussion

The purpose of this study was to determine the effect of TU-Breath Trainer device version III on the cardiorespiratory performance (inspiratory muscle strength and functional capacity) in patients undergoing open heart surgery. The primary finding of this randomized controlled trail is that the group that received inspiratory muscle training combined with cardiac rehabilitation program phase I (active exercise and early ambulation) experienced a more effective recovery of inspiratory muscle strength and functional capacity. Previous studies have demonstrated that IMT can significantly improve functional capacity, particularly in patients with cardiopulmonary conditions and those recovering from cardiac surgery [14,15,17,18,19]. Most studies utilizing IMT have demonstrated its effectiveness in improving inspiratory muscle strength and endurance. IMT is commonly performed using threshold loading devices such as the POWERbreathe electronic device [12,17,18,19] and the spring-loaded Threshold Inspiratory Muscle Trainer [9,10,11], in accordance with the recommendations of the American Thoracic Society and the European Respiratory Society [20]. Moreover, threshold loading devices and electronic devices such as the POWERbreathe device incorporate mouthpieces, which tend to engage accessory respiratory muscles, including the scalene, sternocleidomastoid, trapezius and pectoralis major muscles. The TU-Breath Trainer device is designed to facilitate deep, slow breathing exercises that promote effective use of the primary respiratory muscles, specifically, the diaphragm and costal muscles, while minimizing the involvement of accessory muscles. In this context, the use of the TU-Breath Trainer device in post-cardiac-surgery patients is considered safe. It enhances inspiratory muscle strength (IMS) by enabling attachment to the lower costal margin, promotes increased lung volume and encourages slow, deep breathing techniques to maximize respiratory efficiency. The device offers clinical benefits and contributes to an improved quality of life. Several studies have investigated the use of the TU-Breath Trainer, highlighting its potential to improve IMS, enhance functional capacity and support recovery across diverse patient populations [21,22,23,24].

CPB triggers a systemic inflammatory response that predominantly affects the lungs. The circulation of blood through the CPB circuit stimulates the release of inflammatory cytokines, which contribute to pulmonary injury. This systemic inflammatory response is characterized by impaired pulmonary function, decreased lung compliance, pulmonary edema, reduced functional residual capacity and increased respiratory effort [25]. Additionally, CABG and valve surgery are commonly associated with pulmonary complications such as atelectasis, pneumonia, pleural effusion, impaired pulmonary ventilation and compromised gas exchange [26,27]. We found that atelectasis (36.7%), pleural effusion (33.3%) and pneumonia (30%) were prevalent among postoperative patients, with radiological findings predominantly indicating involvement of the lower lobes. Atelectasis is associated with reduced lung volumes, altered chest wall mechanics and decreased lung compliance. Pulmonary dysfunction increases the effort required for breathing and is linked to impaired chest wall compliance and reduced rib cage mobility, ultimately leading to a decrease in MIP. Diaphragmatic dysfunction and weakness are commonly associated with phrenic nerve injury following cardiac surgery [28].

Prolonged hospital stays are associated with postoperative pulmonary complications and increased mortality. These complications, in turn, elevate the risk of hospital readmission. In this context, the recovery of IMS may help reduce pulmonary complications. A reduction in these complications is also associated with a shorter length of hospital stay following the use of IMT. In this study, the mean length of hospital stay was 8.73 days in the intervention group, compared to 11.7 days in the control group. This finding is consistent with the study by Sabate et al., who reported that an increase in pulmonary complications was associated with a longer postoperative hospital stay [29].

In this study, improvements in IMT were demonstrated by significant increases in MIP and 6MWT performance. During the postoperative period, MIP improved significantly in the group that participated in IMT using a load set at 40% of their baseline MIP. Postoperative IMT improves IMS, enhances endurance and 6MWT performance and reduces length of hospital stay. Zhang et al. [30], in their study in 2023, a literature review of the effects of IMT on patients undergoing CABG, found in patients who underwent CABG improved IMS and endurance, enhanced pulmonary function and increased 6MWT performance. IMT enhances blood flow to the limb muscles, promotes muscle oxygenation and reduces lactate production in the respiratory muscles. The significant increase in MIP observed in the intervention group from after surgery to hospital discharge can be attributed to the effort generated using the TU-Breath Trainer device, which facilitated motor unit recruitment and contributed to gains in muscle strength.

In addition, IMT improved functional capacity, as evidenced by the greater 6MWT distance achieved by the intervention group. A mean difference of 200 m in the 6MWT distance was observed between the control group and the intervention group. Consistent with previous studies, IMT increased exercise capacity in our study, with the greatest improvement observed in the intervention group [31].

This study has some limitations. First, we did not evaluate pulmonary function or assess incision-related pain. Second, we did not analyze variables such as quality of life or levels of anxiety and depression in the patients. Therefore, future studies should include a comprehensive analysis of these variables, along with pain assessment and pulmonary function assessment. In addition, training parameters such as intensity, frequency and duration should be carefully considered, as they may influence the study outcomes.

## 5. Conclusions

IMT can be performed using respiratory device prototypes such as the TU-Breath Trainer and serves as an important strategy for enhancing IMS and cardiorespiratory performance after cardiac surgery.

## Figures and Tables

**Figure 1 arm-93-00010-f001:**
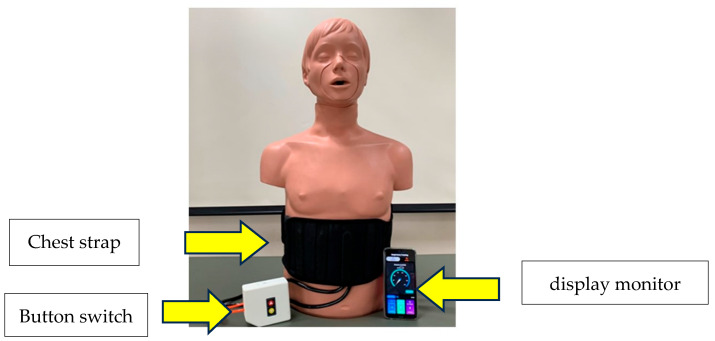
TU-Breath Trainer version III. TU—Thammasat University.

**Figure 2 arm-93-00010-f002:**
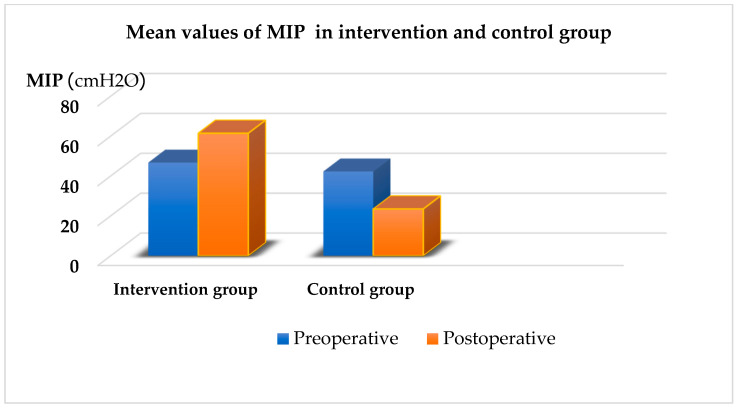
Mean values of MIP in intervention and control group. MIP—maximal inspiratory pressure.

**Figure 3 arm-93-00010-f003:**
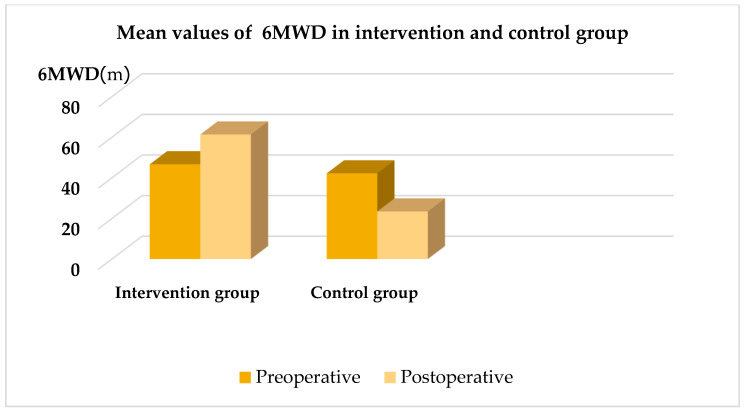
Mean values of 6MWD in intervention and control group. 6MWD—the 6 min walk distance.

**Table 1 arm-93-00010-t001:** Baselines characterize data.

	IG (N = 29)	CG (N = 29)	*p*-Value
Demographic characteristics			
Sex, Female/male	9/20	13/16	0.29
Age (year)	61 (8.61)	64.83 (8.08)	0.62
BMI (kg/m^2^)	24.72 (4.13)	24.25 (4.9)	0.36
Risk factors			
Tobacco smoking	8 (26.7)	11 (36.7)	0.40
Hypertension	24 (80)	24 (80)	0.71
Diabetes	17 (56.7)	21 (70)	0.28
Dyslipidemia	20 (66.7)	21 (70)	0.78
LVEF < 35(%)	7 (23.3)	9 (30)	0.56
Time of surgery (min)	323.67 (90.4)	390 (137.6)	0.04 ^b^
CPB time (min)	109.4 (45.1)	120.6 (48.7)	0.39
Prolonged ventilation (min)	720.9 (274.3)	891.4 (610.5)	0.04 ^b^
Length of stay (day)	8.73 (3.69)	11.7 (6.9)	0.01 ^b^

LVEF—left ventricular ejection fraction. BMI—Body Mass Index. CPB—cardiopulmonary bypass. IG—intervention group. CG—control group. ^b^ Independent *t*-test; significance level *p* < 0.05.

**Table 2 arm-93-00010-t002:** Pulmonary postoperative complications.

Pulmonary Complications	IG (%)	CG (%)	*p*-Value
Atelectasis	2 (6.7)	11 (36.7)	<0.001
Pleural effusion	1 (3.3)	10 (33.3)	<0.001
Pneumonia	0 (0)	9 (30)	<0.001
No complication	27 (90)	0 (0)	<0.001

IG—intervention group. CG—control group.

## Data Availability

Data available on request.

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
