# Peer review of "Inspiratory Muscle Training Improved Cardiorespiratory Performance in Patients Undergoing Open Heart Surgery: A Randomized Controlled Trial"

_arm, 2025, doi:10.3390/arm93030010_

Round 1
Reviewer 1 Report
Comments and Suggestions for Authors
Good paper with clear methodology and design, well writen.
Results are clear ans suported by data. Nevertheless difference between group in term of duration of surgery hospital stay may have influenced results in contril group and worsened the outcomes in this population. This point should be better discussed in term of potential biases.
The type of surgery in each group is also not described.
Author Response
Comments | Response and Revisions |
Good paper with clear methodology and design, well writen.
Results are clear ans suported by data. Nevertheless difference between group in term of duration of surgery hospital stay may have influenced results in contril group and worsened the outcomes in this population. This point should be better discussed in term of potential biases. |
Thank you for pointing this out. We agree with this comment. Therefore, Further details regarding the length of hospital stay (LOS) have been provided: Prolonged hospital stays are associated with postoperative pulmonary complications and increased mortality. These complications, in turn, elevate the risk of hospital readmission. In this context, the recovery of IMS may help reduce pulmonary complications. A reduction in these complications is also associated with a shorter length of hospital stay following the use of IMT. In this study, the mean length of hospital stay was 8.73 days in the intervention group, compared to 11.7 days in the control group. This finding is consistent with the study by Sabate et al., who reported that an increase in pulmonary complications was associated with a longer postoperative hospital stay. Mention in the revised manuscript this change can be found – page number7-8 and line 269-276 |
The type of surgery in each group is also not described. |
Thank you for pointing this out. We agree with this comment. Therefore, supplementary details concerning the types of surgical procedures performed have been incorporated. Mention in the revised manuscript this change can be found – page number4and line 167-168 |
Reviewer 2 Report
Comments and Suggestions for Authors
The authors evaluated the efficacy of inspiratory muscle training on cardiorespiratory performance in patients undergoing open-heart surgery. This topic is worthy of investigation. However, several points need attention before the manuscript can be considered for publication:
- Throughout the manuscript (e.g., lines 22, 25, 26, 30, 39), the term “inspiratory” should be written with a lowercase “i.”
- At first mention of abbreviations (e.g., lines 31 and 77), please provide the full term followed by the abbreviation in parentheses.
- The sentence on line 100 appears incomplete or interrupted; please revise for clarity.
- Please clarify whether the study was blinded, specifying if participants and/or researchers were blinded.
- Explicitly mention the type of study (e.g., randomized controlled trial, observational study) in the Methods section.
- Indicate whether any follow-up assessments were conducted after the intervention.
- Given that patients may have undergone different types of surgeries, please address how surgical variability was controlled or accounted for in the analysis.
- Provide detailed inclusion and exclusion criteria for participant selection.
- Explain the rationale for selecting the age range of 35-80 years for the study population.
- Clarify whether a placebo was used during the study.
- Specify the setting(s) where procedures took place (e.g., inpatient/outpatient).
- Indicate if all surgeries were performed by the same surgeon or different surgeons, and discuss any implications.
- Describe whether patients received the same anesthetic protocols during surgery, including the type and amount, or explain any variability.
- State the qualifications and specialties of the personnel who conducted the experiments, and confirm if they were appropriately trained for the assessments performed.
- Include the significance level (p-value threshold) used for statistical analyses in the Methods section.
- Provide complete definitions of all abbreviations used in figures and tables within their respective footnotes.
- The Discussion section requires reorganization to enhance cohesion and clarity. Please improve logical flow and linkage between ideas.
- The manuscript contains several grammatical errors; a thorough English language review and proofreading are recommended to improve overall readability.
I appreciate your efforts and wish you success in revising the manuscript.
Author Response
Comments | Response and Revisions |
1.Throughout the manuscript (e.g., lines 22, 25, 26, 30, 39), the term “inspiratory” should be written with a lowercase “i.” | Thank you for pointing this out. We agree with this comment. Therefore, we have corrected the 'Inspiratory' to 'inspiratory. Mention in the revised manuscript this change can be found – page number1 and line 22,25,26,30,39. |
2. At first mention of abbreviations (e.g., lines 31 and 77), please provide the full term followed by the abbreviation in parentheses. | Thank you for pointing this out. We agree with this comment. Therefore, we have corrected the “TU breath trainer” to Thammasat University (TU). Mention in the revised manuscript this change can be found – page number1-2 and line31,77. |
3. The sentence on line 100 appears incomplete or interrupted; please revise for clarity |
Thank you for pointing this out. We agree with this comment. Therefore, we complete the sentence as follows: Prior to commencing the training program, participants received a detailed explanation of the study’s objectives and methodology. Mention in the revised manuscript this change can be found – page number3, section2.1. The Study Design and participants and line 100. |
4. Please clarify whether the study was blinded, specifying if participants and/or researchers were blinded. |
Thank you for pointing this out. We agree with this comment. Therefore, we updated and corrected accordingly. In a single-blind study, participants are unaware of whether they have been assigned to the treatment group or the control group. Mention in the revised manuscript this change can be found – page number2, section2.1. The Study Design and participants and line 89. |
5. Explicitly mention the type of study (e.g., randomized controlled trial, observational study) in the Methods section. |
Thank you for pointing this out. We agree with this comment. Therefore, we updated and corrected accordingly. This study was designed as a prospective, single-blind, randomized controlled trial conducted at Thammasat University Hospital. Mention in the revised manuscript this change can be found – page number2, section2.1. The Study Design and participants and line 90. |
6. Indicate whether any follow-up assessments were conducted after the intervention. |
Thank you for pointing this out. We agree with this comment. Therefore, Our study included two assessment time points: one preoperatively and one prior to discharge |
7. Given that patients may have undergone different types of surgeries, please address how surgical variability was controlled or accounted for in the analysis. | Thank you for pointing this out. We agree with this comment. Therefore, we assumed that the type of surgery did not influence the measured parameters, as all patients underwent a median sternotomy. |
8. Provide detailed inclusion and exclusion criteria for participant selection. |
Thank you for pointing this out. We agree with this comment. Therefore, we updated and corrected accordingly. Inclusion criteria for this study were: (1) the patients aged 35-80 years, (2) both men and women, (3) patients were scheduled to undergo cardiac surgery (CABG, mitral valve, aortic valve and atrial septal defect correction). Exclusion criteria for this study were: (1) patients who were unable to understand the techniques, (2) patients with a history of neuromuscular disorder, (3) psychiatric problems, (4) thoracotomy, (5) un-stable angina, (6) uncontrolled cardiac arrhythmia, (7) hemodynamic instability, (8) high resting heart rate (≥ 120 beats per minute), (9) uncontrollable blood pressure or blood glucose. Mention in the revised manuscript this change can be found – page number3, section2.1. The Study Design and participants and line 94-102. |
9. Explain the rationale for selecting the age range of 35-80 years for the study population. |
Thank you for pointing this out. We agree with this comment. Therefore, this age range was selected based on the study by Cargnin et al. 2019, which included a similar population, thereby ensuring comparability with existing research |
10. Clarify whether a placebo was used during the study. |
Thank you for pointing this out. We agree with this comment. This study did not include a placebo group. All participants received the active intervention, as the primary objective was to evaluate its effects without comparison to a placebo." |
11. Specify the setting(s) where procedures took place (e.g., inpatient/outpatient). |
Thank you for pointing this out. We agree with this comment. Therefore, we updated and corrected accordingly. Two physiotherapists recruited the patients, who were admitted to the department of cardiothoracic surgery, Thammasat university hospital. Mention in the revised manuscript this change can be found – page number3, section2.3. procedures and line 128-129. |
12. Indicate if all surgeries were performed by the same surgeon or different surgeons, and discuss any implications. |
Thank you for pointing this out. We agree with this comment. Therefore, all surgeries were performed by the same surgeon. This consistency minimizes variability in surgical technique, which is critical for ensuring uniformity in operative approach and outcomes. |
13. Describe whether patients received the same anesthetic protocols during surgery, including the type and amount, or explain any variability. |
Thank you for pointing this out. We agree with this comment. Therefore, we updated and corrected accordingly. All patients received general anesthesia and underwent median sternotomy. Patients were provided with supplemental oxygen to maintain arterial oxygen saturation above 95% following extubation. Mention in the revised manuscript this change can be found – page number3, section2.3. procedures and line 131-132. |
14. State the qualifications and specialties of the personnel who conducted the experiments, and confirm if they were appropriately trained for the assessments performed. |
Thank you for pointing this out. We agree with this comment. Therefore, we explain more about the physical therapist experience: The physical therapist had over five years of experience in providing care to patients recovering from cardiac surgery. All patients were assessed by the physical therapist both preoperatively and at discharge. Mention in the revised manuscript this change can be found – page number3, section2.3. procedures and line 129-131 |
15. Include the significance level (p-value threshold) used for statistical analyses in the Methods section. |
Thank you for pointing this out. We agree with this comment. Therefore, we updated and corrected accordingly. A p-value of less than 0.05 was considered statistically significant. Mention in the revised manuscript this change can be found – page number4, section2.4. Statistical Analysis and line 160-161 |
16. Provide complete definitions of all abbreviations used in figures and tables within their respective footnotes. |
Thank you for pointing this out. We agree with this comment. Therefore, we have updated and corrected the definitions of all abbreviations used in the figures and tables. Mention in the revised manuscript this change can be found – page number4-6, line 151,209,221,224 |
17. The Discussion section requires reorganization to enhance cohesion and clarity. Please improve logical flow and linkage between ideas. |
Thank you for pointing this out. We agree with this comment. Therefore, we have therefore revised and rewritten the entire discussion section to improve clarity, coherence, and alignment with our study findings. Mention in the revised manuscript this change can be found – page number7-8, line 227-296 |
18. The manuscript contains several grammatical errors; a thorough English language review and proofreading are recommended to improve overall readability. |
Thank you for pointing this out. We agree with this comment. Therefore, we are working to improve the grammar throughout the entire manuscript. |